# Prenatal Opioid and Alcohol Exposures: Association with Altered Placental Serotonin Transporter Structure and/or Expression

**DOI:** 10.3390/ijms252111570

**Published:** 2024-10-28

**Authors:** Nune Darbinian, Nana Merabova, Gabriel Tatevosian, Sandra Adele, Armine Darbinyan, Mary F. Morrison, C. Lindsay DeVane, Sammanda Ramamoorthy, Laura Goetzl, Michael E. Selzer

**Affiliations:** 1Center for Neural Repair and Rehabilitation (Shriners Hospitals Pediatric Research Center), Lewis Katz School of Medicine, Temple University, Philadelphia, PA 19140, USA; nmerabova@gmail.com (N.M.); dr.tatevosian@gmail.com (G.T.); sandra.adele@ndm.ox.ac.uk (S.A.); 2Medical College of Wisconsin-Prevea Health, Green Bay, WI 54304, USA; 3Peter Medawar Building for Pathogen Research, University of Oxford, Oxford OX1 3SY, UK; 4Department of Pathology, Yale University School of Medicine, New Haven, CT 06520, USA; armine.darbinyan@yale.edu; 5Center for Substance Abuse Research, Lewis Katz School of Medicine, Temple University, Philadelphia, PA 19140, USA; mary.morrison@tuhs.temple.edu; 6Department of Psychiatry, Lewis Katz School of Medicine, Temple University, Philadelphia, PA 19140, USA; 7Department of Psychiatry, Medical University of South Carolina, Charleston, SC 29425, USA; devanel@musc.edu; 8Department of Pharmacology and Toxicology, Virginia Commonwealth University, Richmond, VA 23298, USA; sramamoorthy@vcu.edu; 9Department of Obstetrics, Gynecology and Reproductive Sciences, McGovern Medical School at The University of Texas Health Science Center at Houston (UTHealth), Houston, TX 77030, USA; laura.goetzl@uth.tmc.edu; 10Department of Neurology, Lewis Katz School of Medicine at Temple University, Philadelphia, PA 19140, USA

**Keywords:** placental transport, FASD, suboxone, subutex, methadone, vesicles, exosomes

## Abstract

Fetal exposures to many drugs of abuse, e.g., opioids and alcohol (EtOH), are associated with adverse neurodevelopmental problems in early childhood, including abnormalities in activity of the serotonin (5HT) transporter (SERT), which transports 5HT across the placenta. Little is known about the effects of these drugs on SERT expression. Pregnant women who used EtOH or opioids were compared to gestational age-matched controls using a structured questionnaire to determine prenatal substance exposure. Following elective pregnancy termination, placental membranous vesicles and exosomes were prepared from first and second trimester human placentas. Changes in EtOH- or opioid-exposed placental SERT expression and modifications were assessed by quantitative western blot. Novel SERT isoforms were sequenced and analyzed. Opioid-exposed but not EtOH-exposed maternal placentas showed SERT cleavage and formation of new SERT fragments (isoforms). Alcohol-exposed cases showed reduced SERT levels. Antibodies to the N-terminal SERT region did not recognize either of the two cleavage products, while antibodies to the central and C-terminal regions recognized both bands. The secondary band seen in the opioid group may represent a hypophosphorylated SERT fragment. These changes in SERT modifications and expression may result in altered fetal brain serotonergic neurotransmission, which could have neurodevelopmental implications.

## 1. Introduction

Exposure of the fetus to some drugs may have severe effects on brain development and postnatal behavior. Alcohol exposure in utero may result in fetal alcohol spectrum disorders (FASD), including hyperactivity, autistic behavior, and dysregulation of mood [1,2]. Exposure to opioids can result in severe withdrawal syndromes and mood disturbances postnatally [3]. This suggests that these drugs may affect serotonergic neurotransmission. The serotonin (5HT) transporter (SERT) actively transports 5HT and amphetamines to the fetus across the placenta. Prenatal exposure to drugs that alter placental SERT expression or activity may also change fetal synaptic 5HT contents and serotonergic neurotransmission and could alter fetal exposure to medications in early gestation. Unfortunately, not much is known about the effects of drugs, including alcohol or opioids, on SERT expression and activity.

The toxic effects of in utero exposure to alcohol on fetal brain development have been discussed extensively, and several biomarkers for early detection of FASD have been proposed [1,4,5,6,7]. It was also demonstrated that prenatal alcohol exposure was associated with downregulation of SERT in the fetal brain [5]. Effects of prenatal alcohol exposure on protein expression in human placenta have also been described [8]. The exposure was associated with significant reductions in placental expression of VEGFR2 and annexin-A4, but increases in expression of several cytokines, including the neurotoxic TNF-α and IL-13. Other studies provide evidence that prenatal alcohol exposure (PAE) may inhibit SERT expression while simultaneously promoting increased tryptophan hydroxylase 1 (TPH1) protein expression in the human placenta [9]. However, not much is known about the role of placental SERT, or its alterations due to maternal alcohol consumption, in the pathogenesis of FASD.

The placenta serves as the key regulator of fetal exposure to maternal use of opioids. Opioisd use during pregnancy has been studied for many years and is associated with neonatal abstinence syndrome. Opioids can cross the placenta and accumulate in the amniotic fluid [10]. Thus, it is important to study the effects of changes in placental transporters on the exposure of fetuses to opioids. Recently, we demonstrated the effects of prenatal opioid exposure on opioid receptors and on expression of fetal brain miRNAs [11], but the effects on placental transporters have not yet been explored.

The placenta serves an important barrier function in protecting the fetus from exposure to maternal drugs and other metabolites. The fetal-maternal-placental circulation is established by the 10th week of gestation [12]. Placental transport is regulated by two layers of the syncytiotrophoblast, the basal membrane on the fetal side, and the apical syncytial microvillous membrane (brush border) on the maternal side [12]. Drugs cross the placenta either through passive diffusion or via active membrane transport. Only hydrophobic, non-ionized molecules less than 600 Da in size cross by passive diffusion [12]. Increased levels of an efflux transporter on the apical surface would decrease fetal exposure to its substrate drug, while increased levels of an efflux transporter on the basolateral surface would increase fetal exposure [13]. The placenta itself has enzymes that can metabolize drugs, and those enzymes are expressed in the first trimester (to protect fetuses from teratogens, including opioids and alcohol), but then their expression declines. Placental transporters can participate in the degradation of drugs and their metabolites by placental enzymes, helping to minimize fetal drug exposure [14].

Some drug exposures may be disastrous to the fetus, but for most drugs, the effects are poorly understood. To evaluate the effects of drugs on fetal development, we recently studied the expressions and activities of placental influx transporters (maternal to fetal) and efflux transporters (fetal to maternal) in normal drug non-exposed placental vesicles during each of the three trimesters [14,15,16].

Given the growing number of women taking opioids, it is important to investigate how these substances affect placental SERT, an influx transporter located in the apical syncytiotrophoblast. SERT is also located in other sites throughout the body, including the blood-brain barrier [14]. The gene encoding SERT is located on chromosome 17q11.2 and is translated to create a 12 transmembrane domain protein, which regulates reuptake of monoamine neurotransmitters [5,17]. Placental SERT regulates the concentration of 5HT in the intervillous space to ensure adequate placental blood flow. SERT in the fetal brain likely regulates levels of monoamine neurotransmitters and thus fetal neurodevelopment [18,19,20]. Many opioids are also substrates for the ATP-binding cassette (ABC) transporter, also known as the permeability glycoprotein (P-gp) transporter or multi-drug resistance protein (MDRP) transporter, encoded by the ABC subfamily B member 1 gene (*ABCB1/P-gp/MDRP1*). In the placenta, this transporter is also found mainly in the cells of the brush border. The epitope for the mAb used in our assay for this transporter is located on the intracellular side of the membrane. Thus, ABCB1 levels were quantified in inside-out placental brush border membrane vesicles (IOV).

Placental vesicles are intracellular vesicles obtained by homogenization of placental tissue. Exosomes are a form of extracellular vesicles, derived from intracellular membrane-bound multivesicular bodies that fuse with the plasma membrane and release their vesicles into the extracellular space. Placental-derived exosomes are exosomes from placental tissue that have entered into the maternal blood, and therefore, can be isolated by sampling maternal serum. Similarly, as we have described previously, fetal brain-derived exosomes are exosomes released by fetal brain cells. They can cross the placenta and enter the maternal blood [6].

In the present study, we focused primarily on defining the alcohol and opioid exposure-associated changes in placental SERT protein expression and modification and to correlate these changes with changes in placenta-derived exosomes isolated from maternal blood. We hypothesized that maternal exposure to opioids and/or alcohol would alter placental levels of SERT compared to gestational age-matched controls. Changes in placental SERT would also suggest that women exposed to opioids or alcohol might have increased fetal exposure to other drugs.

## 2. Results

### 2.1. Opioid Exposure Is Associated with a Decrease in Expression of SERT in Placental Vesicles

Using an antibody that binds to the extracellular domain of SERT, we previously found that SERT protein levels increased with GA in right-side-out placental vesicles (ROVs) isolated from non-substance-using mothers. We also identified a heretofore undescribed isoform of the 70 kDa SERT that was only 34 kDa and was prominent in first trimester samples [15]. In the present study, we compared maternal blood and placental tissues from 20 opioid-exposed, 20 EtOH-exposed, and 11 neuroactive medication-exposed fetuses from late first and second trimester pregnancies with each other and with 20 GA-matched controls. The clinical characteristics of the subjects are presented in Table 1. Quantitative western (qWestern) blot analysis was performed on protein lysates prepared from the placentas. Levels of SERT protein were reduced in the placentas of women who used opioids during pregnancy, compared with unexposed controls (Figure 1A).

### 2.2. Opioid Exposure Is Associated with Reduced Placental Expression of ABCB1

Since many opioids are ABCB1 substrates, ABCB1 protein levels were also quantified in IOV by Western blot assays. Levels of ABCB1 were reduced in opioid cases (Figure 1B).

### 2.3. Maternal Opioid Use Is Associated with SERT Modification

Subjects who admitted to chronic opioid use during pregnancy were compared to GA-matched controls and to subjects admitting to use of other drugs, such as SSRIs (Figure 2A, lanes 5–8), amphetamines, or the anti-epileptic drug levetiracetam (Keppra, Figure 2B). Only maternal opioid exposure was associated with alteration of the SERT protein band pattern (lanes 4–10). Lighter exposure (middle panel) and higher magnification showed two additional bands representing cleaved forms of SERT in opioid cases only (Figure 2C). Thus, maternal opioid exposure was associated with alteration of the SERT protein band pattern, with novel cleavage-derived isoforms of SERT present only in opioid-exposed cases. The cleavage appeared to be dose-dependent. The strongest SERT cleavage was in second trimester cases compared to first trimester. Methadone exposure plus high BMI, or exposure to Suboxone or Subutex, even if BMI was normal, also was associated with intense SERT cleavage. SERT cleavage was greater at methadone doses of 135 or 75 mg/day (taken over a period of 2 months) than at 45 mg/day. Interestingly, a patient who had been taking 140 mg/day of methadone 6 months previously but more recently used only 54 mg/day had low SERT cleavage levels. With suboxone use, the strongest cleavage was seen at a dose of 8 mg/day (started 3 weeks previously, during the first trimester), and the lowest amount of cleavage was at 1–2 mg per day. These observations suggest that opioid-induced SERT cleavage may be a relatively acute and reversible process.

### 2.4. The Sequences of Opioid-Associated Cleaved SERT Isoforms Are from the Central Domain

Anti-SERT antibodies were used to detect the central and C-terminal domains of SERT in 12 vesicles from first and second trimester opioid-exposed placentas (Figure 3). Treatment with phosphatase inhibitors (Figure 3A, lane 5 vs. lane 6) resulted in enhancement of the lower double band, consistent with the fact that the epitope for the antibody used in the assay contains a phosphorylation site (Thr276 of the complete SERT sequence) and that the antibody binds optimally to the unphosphorylated form. This suggests that the secondary band seen in the opioid group may represent a dephosphorylated SERT isoform. Thus, exposure to opioids may not only increase SERT instability but also may enhance vulnerability to dephosphorylation. Antibodies specific to the N-terminal epitope of SERT did not recognize either of the two smaller (32/34 kDa) bands, while an antibody to the C-terminal region recognized both bands (Figure 3A, upper panel), which had the same MW as the bands identified by antibodies to the central region of SERT (lower panel). Therefore, the novel SERT isoforms in opioid cases represent central and C-terminal SERT fragments.

The purity of placental vesicles, the specificity of the SERT double bands, and the lack of contamination of the placental vesicle fraction were confirmed using an antibody specific to the placental 110 kDA macrophage surface marker CD163. No 110 kDa band was found (Figure 3B), suggesting that the placental vesicles were not contaminated by placental tissue. We previously tested placental vesicles isolated with our methods for the absence of Contactin 2/Tag1, a fetal neuron-specific marker that we use to isolate fetal brain-derived exosomes (FB-Es). We also confirmed the presence of alkaline phosphatase (ALP) in placental vesicles. Absence of ALP in FB-Es was used as an additional indicator of the purity of the placental vesicles [7].

### 2.5. SERT Sequencing in Opioid-Exposed Placentas: SERT Isoforms Are Cleaved at the Thr276 Phosphorylation and Activation Site

SERT sequencing of novel bands in opioid cases was performed by the Edman degradation experiment (Figure 4) to get the N-terminal sequence of truncated SERT protein double bands (shown in Figure 2 and Figure 3). Two SERT fragments were present in the opioid samples. The sequences were compared with two 5HT transporter isoforms downloaded from Unipro. Sequence alignment was performed using ClustalW2 on two isoforms of SERT fragments on SLC6A4 (synonyms: HTT, SERT; Organism Homo sapiens (Human), Taxonomic identifier 9606 [NCBI]). The first four amino acids of the central region of SERT were detected in two truncated bands corresponding to the sequence of SERT. In combination with the western blot results, this confirmed the presence of truncated SERT protein. Using the sequence alignment tool, we located truncated proteins. Version 1 (the upper band) extends from amino acid 183–227 (out of 672) of human SERT. Truncated Version 2 (lower band) extends from amino acid 224–276 of human SERT and includes the phosphorylation site Thr276.

### 2.6. Prenatal Alcohol Exposure Was Associated with Downregulation of SERT and ABCB1 in Placental Vesicles and Placenta-Derived Exosomes

We compared the effects of EtOH exposure on SERT downregulation in placental vesicles with the effects of opioid exposure. We also compared the levels of SERT in placenta-derived exosomes (PEs) with previously studied SERT levels in fetal brain-derived exosomes [5]. Prenatal EtOH exposure was associated with a 60% reduction of total SERT in brush border membranous vesicles (Figure 5A, bars 1–2) and a 62% reduction of SERT in placenta-derived exosomes (bars 3–4) compared to unexposed controls. However, unlike the vesicles from opioid-exposed placentas, vesicles of EtOH-exposed placentas did not show evidence for an increase in SERT cleavage and the formation of higher levels of small SERT isoforms. Similarly, EtOH use by the mother during pregnancy was associated with reduced expression of ABCB1 in IOV and PEs (Figure 5B) and reduced SERT levels in FB-Es.

## 3. Discussion

The present results show that maternal use of both opioids and EtOH is associated with reductions in SERT levels in placental vesicles and in FB-Es, but by different mechanisms. Whereas opioids were associated with SERT cleavage and generation of small isoforms, EtOH was associated only with reduced SERT protein levels. Others have shown that inhibiting SERT activity may contribute to serotonergic effects when opioids are given in combination with other drugs, e.g., monoamine oxidase inhibitors or selective 5HT reuptake inhibitors [21].

### 3.1. The 5HT Transporter (SERT) Isoforms

Two novel effects of maternal opioid use on placental SERT content were observed. First, there was reduced SERT content in opioid-exposed placentas. Second, there were increased amounts of SERT cleavage products in the placentas, and a novel SERT isoform was present only in opioid-exposed cases. In addition to the 70 kDa SERT protein, in opioid-exposed placentas, there were two truncated C-terminal SERT proteins in the 32–34 kDa range. The cleavage products were sequenced, but their possible functions are not known. One of the secondary bands seen in the opioid group contains the phosphorylation site Thr276. It probably exists mostly in the unphosphorylated state, which may bind poorly to the antibody used in the Western blot since this band became denser when alkaline phosphatase was inhibited. If opioids have a similar effect on full-length SERT, this also might act to decrease the transport activity by any remaining uncleaved SERT. Previous work in the laboratory of one of our authors showed that psychostimulants prevented phosphorylation and induced sequestration of 5HT transporters [22]. The possible effects of this on fetal development are not yet known.

### 3.2. Transporter Expression and Activation

The above suggests that the opioid-associated reduction in placental vesicle SERT levels may be the result of increased cleavage rather than, or in addition to, decreased production. Whether increased SERT cleavage in opioid cases leads to reductions in SERT activity during gestation is not known. Maternal opioid exposure reduced total placental SERT protein levels early in gestation, which may result in loss of transporter activity and might alter fetal exposure to other medications. Sequencing of the two SERT fragments (cleaved SERT; clSERT) in opioid-exposed cases confirmed the presence of a threonine site (Thr-276) in the smaller of the two isoforms, which could be phosphorylated by cyclic guanosine monophosphate (cGMP)-dependent protein kinase G (PKG) [23], thus activating SERT. It is not known whether clSERT is active as a transporter. If it is, then its hypophosphorylation might also contribute to the overall suppression of SERT activity in the placenta. Myristylation of PKGII (PKG isoform II) blocks its ability to activate SERT, possibly by anchoring PKGII to the membrane [24]. The small isoform of SERT also contains two glycosylation sites, which are required for the homo-oligomerization involved in transport function. Finally, phosphorylation of a tyrosine moiety in SERT supports transporter protein stability and 5HT transport [25].

The changes in placental SERT described here may prove valuable as markers for altered fetal brain 5HT neurotransmission, which might affect neurodevelopment.

### 3.3. Placental ABCB1 Changes upon Opioid Exposure

ABCB1 restricts the transfer of drugs and other alien substances, such as oncogenic toxins, from the mother to the fetus by transporting them from the cells of the placenta back to the maternal circulation. ABCB1 serves an important role in shielding the fetus from teratogenic substances in early gestation, thereby reducing the incidence of congenital anomalies [26]. Transport of ABCB1 substrates is more than 100 times greater from fetus to mother than transport from mother to fetus. However, applying ABCB1 inhibitors to the fetal side had little effect on this transport [26]. This probably means that most of the placental ABCB1 is located in the apical cells of the placental trophoblast, the brush border. The present data, along with those presented previously [15], indicate that opioid exposure reduces placental ABCB1 expression during all three trimesters.

In the present study, prenatal alcohol exposure was also associated with reduced ABCB1 levels in human placental IOVs and PEs isolated from maternal blood. This is consistent with findings that previously had been obtained in male alcohol-preferring rats [27].

### 3.4. Biomarkers for FASD

FASD has been subdivided into several syndromes [1], of which the most severe, fetal alcohol syndrome (FAS), includes the full spectrum of craniofacial, somatic, and neurobehavioral abnormalities. We previously demonstrated that molecular abnormalities are already present in fetuses exposed to EtOH [5]. These molecular abnormalities might reflect a direct effect of EtOH on the developing brain, or they might reflect transplacental movement of molecules associated with comorbid conditions in the mother. For example, maternal depression is seen frequently in pregnant women who drink EtOH. These women may take antidepressant drugs such as serotonin-specific reuptake inhibitors (SSRIs). Our previous data suggested that maternal EtOH consumption, maternal depression, and maternal SSRI use, each independently and in combination, are associated with abnormalities at the mRNA and protein levels of molecular pathways (i.e., 5HT and dopamine) that are associated with abnormal mood regulation in adult humans. Among these abnormalities was a dramatic downregulation of SERT expression in fetal brain and FB-E. In the present study, exposure to EtOH, SSRIs, or amphetamines was associated with downregulation of SERT in both the placental membranous vesicles and PEs. Thus, the EtOH-associated changes in placenta appear to be similar to those in the fetal brain and may reflect molecular abnormalities that contribute to the pathogenesis of FASD.

### 3.5. Differential Effects of Opioids vs. SSRI or Amphetamines on SERT Cleavage

Placental SERT levels are reduced progressively throughout normal fetal development in humans [15]. In the present study, opioid but not SSRI or amphetamine exposure was associated with SERT cleavage and formation of novel SERT isoforms. Thus, early detection of novel SERT isoforms could serve as a biomarker to predict the emergence of opioid abstinence syndrome, while low SERT levels without cleavage products might serve as an early biomarker for FASD.

### 3.6. Placenta-Derived Exosomes (PEs)

Previously, we developed a non-invasive method to investigate fetal brain proteins and RNAs by isolating FB-Es from maternal serum [4,5,6,7]. In the present study, this strategy was further developed to investigate SERT in the placenta and in PEs. Maternal serum proteins have been used to predict infant outcomes and might be useful in classifying difficult-to-diagnose FASD subpopulations. However, the ability to isolate PEs non-invasively from maternal blood and analyze their cargos, e.g., for SERT and its isoforms, as early as the first trimester, might prove more specific in identifying molecular markers to predict the emergence of neurodevelopmental abnormalities such as FASD and opioid abstinence syndrome in at-risk children.

### 3.7. Limitations

The present study used fetal tissues and maternal blood samples from mothers who elected to terminate their pregnancies. This gave rise to two important limitations. First, it limited the number of fetuses that could be studied. In addition, it did not allow for postnatal follow-up to determine the clinical outcomes in the offspring. We tried to compensate for the first limitation by using a case-control design, matching each substance-exposed fetus with one of equal gestational age, sex, and maternal age, but this was not sufficient to control for several other factors such as maternal metabolic status, tobacco use, race, and socioeconomic status. In addition, it is obviously impossible ethically to perform a prospective study assigning some mothers to use substances and others not. Another limitation due to the absence of prospective recruitment of mothers is that we had to rely on self-report to determine use or abstinence with regard to drug use. In studying some other drugs of abuse in our patient population, there was a high correlation between self-report and drug test results. Moreover, any underreporting of such use would have the effect of reducing rather than exaggerating the reported effects of substance exposure. Thus, the predictive value of any biomarker discovered in this study would be at least as great, if not greater, than the reported results would suggest. A much larger study on FASD is underway, in which mothers are recruited prospectively, blood and urine tests will verify EtOH use, and the pregnancies are not interrupted. The children will be followed up to determine which biomarkers best predict whether an at-risk child will go on to have one of the forms of FASD. A similar approach might be used for opioid or other toxic exposures.

## 4. Materials and Methods

### 4.1. Clinical Recruitment

First and second trimester placental tissue and maternal serum were collected from women undergoing elective pregnancy termination under Temple University IRB-approved protocols (#20798: Placental Transport of Psychoactive Drugs Across Gestation, PI Dr. Goetzl, Laura; #21476: Early Gestation Alcohol Exposure: Mechanisms of Human Developmental Injury, PI Dr. Darbinian, Nune). A face-to-face history was conducted by a trained study coordinator, and an extensive medication and substance exposure history was obtained. The amount of EtOH was calculated as the total number of drinks consumed in a week multiplied by the number of weeks of exposure. A detailed questionnaire was used based on the NICHD PASS study. Each drink was estimated as the equivalent of one shot (1.5 oz of brandy or 5 oz of wine).

First trimester samples were collected between 8 and 13 weeks GA, and second trimester samples between 14 and 23 weeks GA. Apical brush border membrane fractions were prepared from fresh placentas. Placenta-derived exosomes were isolated from maternal serum. Maternal questionnaires were used to determine prenatal exposure to medications, drugs, and alcohol. Twenty subjects who admitted to chronic alcohol and 20 who took opioids (methadone, suboxone, and oxycodone) were compared to 20 unexposed controls matched individually by GA, fetal sex, and maternal age. For some comparisons, a separate group of 11 mothers taking other neuroactive drugs (amphetamines, SSRIs, or antiepileptic drugs) were included. These also were compared with matched controls. Placental tissue and maternal blood samples were analyzed in the laboratory as summarized in Table 2.

### 4.2. Preparation of Brush Border Membrane Vesicles

Fresh placenta tissue was immediately rinsed in cold phosphate-buffered saline (PBS) to remove any blood. Villous tissue was placed on ice and transported immediately from the clinical setting to the laboratory (less than 1 h). Apical brush border membrane fractions were prepared according to our published protocol [15], modified slightly from [28]. The purity of brush border membrane preparations was assessed by the enrichment of alkaline phosphatase (ALP) activity using a colorimetric ALP Assay Kit (Abcam, Cambridge, UK) according to the manufacturer’s instruction. Optical density at 405 nm was measured using an ELx808 microplate reader (Bio-Tec Inc., Winooski, VT, USA) and Gen5 software (BioTek Gen5 v2.09 Data Collection Analysis, Winooski, VT, USA).

### 4.3. IOV Preparation

Plasma membrane preparations of high purity (about 95%) were obtained by partitioning in aqueous polymer two-phase systems. These preparations contain right-side-out vesicles (ROV). Part of these vesicles have been turned inside-out by freezing and thawing, and inside-out and right-side-out vesicles subsequently were separated by repeating the phase partition step. The increased number of freeze/thaw cycles significantly increased the yield of IOVs. ATPase activity was determined. Freezing and thawing of plasma membranes has been reported to cause increased ATPase activity by “unmasking” of latent ATP binding sites [29]. In the ATPase assay, lamellae and IOVs contribute to the activity, while ROVs do not, since the ATP-binding domain of transporters is located intravesicularly. ALP assay was performed for ROV [27].

### 4.4. Isolation of Placenta-Derived Exosomes (PEs) from Maternal Serum and ELISA Quantification of Placenta SERT Protein

Human PEs were isolated using 250 mL of serum and incubated with cocktails of protease and phosphatase inhibitors. After incubation, serum was further incubated with exosome precipitation solution (EXOQ; System Biosciences, Inc., Mountainview, CA, USA). All further steps were completed according to our previously published protocols [6]. To isolate exosomes from placental sources, total exosome suspensions were incubated for 90 min at 20 °C with 50 mL of 3% bovine serum albumin (BSA) (Thermo Scientific, Inc., Waltham, MA, USA) containing 2 mg of mouse monoclonal IgG1 antihuman ALP antibody that had been biotinylated (EZLink sulfo-NHS-biotin System, Thermo Scientific, Inc., Waltham, MA, USA). Exosome isolation concluded as previously described [6]. For exosome counts, immunoprecipitated pellets were resuspended in 0.25 mL of 0.05 mol/L glycine-HCl (pH 3.0) at 4 °C with pH 7.0 with 1 mol/L Tris-HCl (pH 8.6). Exosome suspensions were diluted 1:200 to permit counting in the range of 1–5 × 10^8^/mL with an NS500 nanoparticle tracking system (NanoSight, Amesbury, UK). Resulted PE (placenta exosomes) were used in ELISA assays.

### 4.5. Alkaline Phosphatase (ALP) Activity

The purity of brush border membrane preparations was assessed by the enrichment of ALP activity. ALP activity was assessed using a colorimetric (spectrophotometric) enzymatic ALP Assay Kit (Abcam) in duplicates according to the manufacturer’s instruction. ALP catalyzes the hydrolysis of phosphate esters in alkaline buffer and produces an organic radical and inorganic phosphate. Optical density at 405 nm was measured using an ELx808 microplate reader (Bio-Tec) and Gen5 software. P-nitrophenyl phosphate (pNPP) was used as a phosphatase substrate, which turns yellow (λmax = 405 nm) when dephosphorylated by ALP. The pNPP standards were prepared and incubated with ALP enzyme for 60 min to generate the standard curve (0–20 nmol/reaction). Samples were incubated with pNPP for 60 min before adding stop solution. Results were determined by comparing the value of the sample to a standard whose value is known by using a standard curve to determine the concentration of the metabolite in each sample. Average enzyme activity was normalized to total protein: ALP activity (U/mL) = pNP generated by the sample (in μmol)/volume of sample in reaction (mL)/reaction time (minutes).

### 4.6. Isolation of Fetal Brain-Derived Exosomes (FB-Es) from Maternal Serum and ELISA Quantification of Exosomal Proteins

Human FB-Es were isolated as described previously [6]. To isolate exosomes from fetal neural sources, total exosome suspensions were incubated for 90 min at 20 °C with 50 μL of 3% bovine serum albumin (BSA; Thermo Scientific, Inc., Waltham, MA, USA) containing 2 μg of mouse monoclonal IgG1 antihuman contactin-2/TAG1 antibody (clone 372913, R&D Systems, Inc., Minneapolis, MN, USA) that had been biotinylated (EZLink sulfo-NHS-biotin System, Thermo Scientific, Inc., Waltham, MA, USA). Then, 10 μL of Streptavidin-Plus UltraLink resin (PierceThermo Scientific Inc., Waltham, MA, USA) in 40 μL of 3% BSA was added, and the incubation continued for 60 min at 20 °C. After centrifugation at 300× *g* for 10 min at 4 °C and removal of supernatants, pellets were resuspended in 75 μL of 0.05 mol/L glycine-HCl (pH 3.0), incubated at 4 °C for 10 min, and recentrifuged at 4000× *g* for 10 min at 4 °C. Each supernatant was mixed in a new 1.5 mL Eppendorf tube with 5 mL of 1 mol/L Tris-HCl (pH 8.0) and 20 μL of 3% BSA, followed by the addition of 0.4 mL of mammalian protein extraction reagent (M-PER; Thermo Scientific Inc., Waltham, MA, USA) containing protease and phosphatase inhibitors prior to storage at −80 °C. For the exosome counts, immunoprecipitated pellets were resuspended in 0.25 mL of 0.05 mol/L glycine-HCl (pH 3.0) at 4 °C with a pH of 7.0 and 1 mol/L Tris-HCl (pH 8.6). The exosome suspensions were diluted 1:200 to permit counting in the range of 1–5 × 10^8^/mL with an NS500 nanoparticle tracking system (NanoSight, Amesbury, UK).

### 4.7. ELISA Quantification of Exosomal Proteins

ELISA, a plate-based assay technique, was used for detecting and quantifying SERT and P-gp transporters in ROV or IOV using a human SERT ELISA kit (NeoBiolab, Cambridge, MA, USA) and a human multidrug resistance protein 1 (P-gP) ELISA kit (Antibody Research, St. Charles, MO, USA). Downregulation of SERT in placenta-derived exosomes was assessed by ELISA. Proteins and CD81 (American Research Products-Cusabio) were quantified according to the suppliers’ instructions. ELISA data were statistically evaluated using Excel (Microsoft 365, software version 2404) and statistical analysis tools: CurveExpert for ELISA statistics (CUSABIO) or APP 96-well Plate Assay Data Analysis Software 5.0.apk (Cloud-Clone, Katy, TX, USA), available online.

### 4.8. Quantitative Western Blot Assays

Changes in opioid-exposed placental SERT expression and modification relative to housekeeping proteins and unexposed controls were measured by quantitative western blotting, as previously described [4,5]. In brief, protein extracts were prepared from apical membrane fractions. Placenta was obtained and processed within 30 min to prepare human placental brush-border membrane vesicles. Tissue was cut into small pieces, washed, and rinsed with wash buffer (10 mM HEPES, 300 mM mannitol, adjust pH with Tris to 7.0, protease inhibitor cocktail 1:100 dilution) three times. The tissue was placed in cold wash buffer and agitated for 30 min at 4 °C. The suspension was filtered again using cheesecloth. The filtrate was centrifuged at 7000 rpm for 15 min. The pellet was discarded, and the supernatants were centrifuged at 25,000 rpm at 4 °C for 30 min. The pellet was washed with cold buffer and homogenized in ice, then samples were placed in wash buffer in a small pre-chilled beaker with a stir bar with MgCl_2_ (10 mM final concentration) for 1 min and incubated on ice for 10 min. The suspension was centrifuged at 4300 rpm for 15 min at 4 °C. The pellet was discarded, and the supernatants were centrifuged again at 25,000 rpm for 30 min at 4 °C. The pellet was homogenized again using a 25-G syringe to make suspension. Samples were transferred into a new centrifuge tube with a buffer, centrifuged for 30 min at 25,000 rpm, and the pellet was resuspended in preloading buffer. An alkaline phosphatase assay was performed, and protein concentration was determined using the Bradford assay. Aliquots were stored in liquid nitrogen for further protein analysis by western-blot assay and transporter activity assay. Cellular debris was removed by centrifugation for 5 min at 4 °C, 14,000 rpm, and the supernatant was assayed for protein content by Bradford analysis (Bio-Rad, Hercules, CA, USA).

Loading dose was determined by protein concentration. Vesicle proteins (30 μg) in Laemmli sample buffer were heated at 95 °C for 10 min, separated by gradient SDS-PAGE, and transferred to a NC membrane. Proteins were detected using specific primary antibodies (anti-SERT, 1:1000 dilution) and secondary IRDye^®^ dyes (IRDye^®^ 800CW Goat Anti-Rabbit and IRDye^®^ 680RD Goat Anti-Mouse Li-COR dyes, 1:10,000) with the Odyssey^®^ CLx Imaging System (LI-COR, Inc., Lincoln, NE, USA). Band intensity (normalized to Grb2, Actin, or GAPDH) was detected, visualized, and quantified using iS Image Studio™ Software version 3.1. Consistent loading of wells was also verified by Coomassie Blue staining.

### 4.9. Antibodies

Anti-Serotonin Transporter rabbit polyclonal antibody/AB10514P (Central region) was purchased from EMD Millipore (Billerica, MA, USA). Anti-SLC6A4 SERT 95 kDa antibody, LS-C156102 (internal), was from LSBio LifeSpan BioSciences, Inc. Anti-Serotonin Transporter (N-terminal) (SAB4200039) was from Sigma (St. Louis, MO, USA). SLC6A4 antibody to C-terminus SERT (OAEB02575) was from Aviva Systems Biology, Corp. (San Diego, CA, USA). Anti-ABCB1 P-gp mouse monoclonal antibody Clone 2F7 PGP) was obtained from OriGene (Rockville, MD, USA). Loading control antibody to GAPDH (6C5, sc-32233) was obtained from Santa Cruz Biotechnologies (Santa Cruz, CA, USA); mouse monoclonal Grb2 was obtained from BD Biosciences (San Jose, CA, USA); anti-α-Tubulin clone B512 was obtained from Sigma-Aldrich (Sigma-Aldrich Co. St. Louis, MO, USA).

### 4.10. SERT Sequencing, Instrument, and Procedure

Novel SERT isoforms in opioid-exposed cases were sequenced and analyzed. An ABI Procise 494 sequencer was used. The Edman degradation chemistry procedure was performed. Sample preparation: for the sequencing of N-terminal residues by Edman degradation, 30 μgm of placental vesicles isolated from first trimester control samples were used for SDS-polyacrylamide gel electrophoresis (SDS-PAGE). Proteins were transferred onto PVDF membranes and stained with Coomassie Blue solution. Membranes were washed with water to remove the glycine. Bands with sizes corresponding to 32–34 kDa MW were cut, and the Edman degradation experiment was performed to get the N-terminal sequence of truncated SERT protein versions (CD Biosciences Inc., Shirley, NY, USA, www.creative-biolabs.com, accessed in 2005). Two potential C-terminal SERT fragments were detected. The sequences were compared with two 5HT transporter isoforms downloaded from Unipro. Sequence alignment was performed using ClustalW2 on two isoforms of SERT and potential N-terminal fragments (http://www.uniprot.org/uniprot/P31645#P31645, 1 October 2014 to 2 October 2024, gene names SLC6A4; synonyms: HTT, SERT; Organism Homo sapiens (Human), Taxonomic identifier9606 [NCBI]). The first four amino acids of the central region of SERT were detected in the upper fragment corresponding to the sequence of SERT; the lower band corresponded to the C-terminal region of SERT. In combination with the western blot results, this confirmed the presence of truncated SERT protein in opioid-exposed placental vesicles.

### 4.11. Statistical Analysis

Statistical analysis was performed using SPSS Statistics from IBM Corp., released in 2017 for Windows, Version 25.0 (Armonk, NY, USA). All data are represented as the mean ± SD for all performed repetitions. Means were analyzed by a one-way ANOVA, with Bonferroni correction, where appropriate. Statistical significance was defined as *p* < 0.05. Sample numbers are indicated in the figure legends.

### 4.12. Ethics: Human Subjects

Consenting mothers were enrolled at between 9 and 23 weeks’ gestation, under a protocol approved by our Institutional Review Board (IRB). This protocol involved no invasive procedures other than routine care. Maternal EtOH exposure was determined with a face-to-face questionnaire that also included questions regarding many types of drugs and medications used [4,5]. The questionnaire was adapted from that designed to identify and quantify maternal EtOH exposure in the NIH/NIAAA Prenatal Alcohol and SIDS and Stillbirth (PASS) study [30].

All procedures involving the collection and processing of blood and placenta tissues were completed according to NIH Guidelines through a trained study coordinator. All investigators were trained annually to complete Citi Program—Human Subject training, Biohazard Waste Safety Training and Blood-Borne Pathogens Training, and all other required training. Written informed consent has been obtained from the patients for studies, and de-identified samples were used for this publication. Informed consent forms were maintained by the study coordinator. The de-identified log sheets contain an assigned accession number, the age, sex, ethnicity, and race of the patient. Except for an assigned accession number, no identification was kept on the blood and placenta samples.

*Eligibility Criteria.* The blood and placenta samples were obtained according to NIH Guidelines through a trained study coordinator. Samples were collected regardless of sex, ethnicity, and race. Subjects were excluded if they had an active urinary tract infection on history, nitrates, or WBCs on clinical UA; no prisoners; no adults who are cognitively impaired or physically unable to provide consent to participate; no patients with severe blood disorders (e.g., hemophilia).

*Treatment Plan.* Each patient was asked to sign a separate consent form for research on blood and tissue samples. Blood obtained was processed for the collection of serum. No invasive procedures were performed on the mother other than those used in her routine medical care. Placenta tissues were processed for protein isolation.

*Risk and Benefits.* There were very small risks of loss of privacy, as with any research study in which protected health information is viewed. The samples were depersonalized before they were sent to the lab for analysis. There were no additional risks of blood sampling, as this was only performed in subjects with clinically indicated venous access. There was little anticipated risk from obtaining 2–3 cc of blood, but a well-trained study coordinator collected all samples.

There was no direct benefit to the research subjects from participation, but there is significant potential benefit for the future FASD subjects and the general population. This research represents a reasonable opportunity to further the understanding, prevention, or alleviation of a serious problem affecting the health or welfare of FASD patients.

*Informed Consent.* Consent forms were maintained by the study coordinator and were not sent to the investigator with the samples. The de-identified log sheets and IRB protocol were sent by the study coordinator to the principal investigator with each blood and tissue sample. This sheet contained an assigned accession number, as well as the age, sex, ethnicity, and race of the patient. Except for the accession number, no identification was kept on the blood and tissue samples.

## Figures and Tables

**Figure 1 ijms-25-11570-f001:**
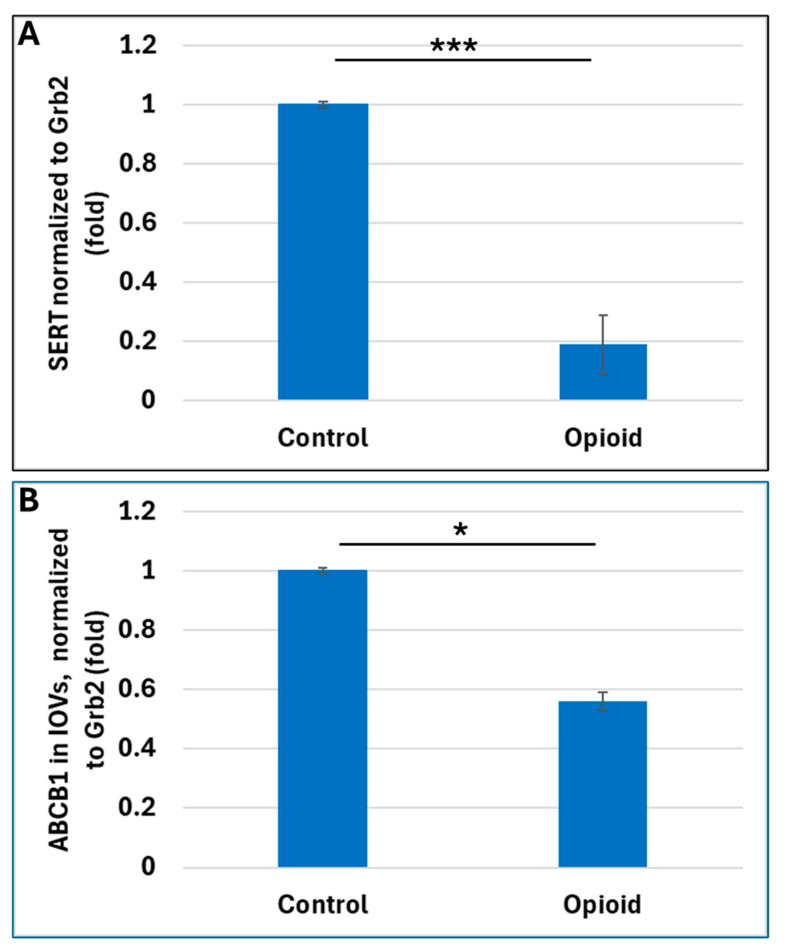
**Effect of *in utero* opioid exposure on SERT and ABCB1 expression in human placental membranous vesicles.** (**A**) Downregulation of SERT levels in qWestern blot analysis of protein lysates, comparing 20 first and second trimester opioid-exposed human placentas with 20 unexposed controls individually matched for fetal sex, GA, and maternal age. (**B**) Reduced levels of the drug transporter ABCB1 in opioid-exposed placentas. Inside out placental brush border membrane vesicles were used to measure and quantify the ABCB1 expression. In both (**A**,**B**), Grb2 served as a loading control. Data are presented as fold change (* for *p* < 0.05, *** for *p* < 0.001).

**Figure 2 ijms-25-11570-f002:**
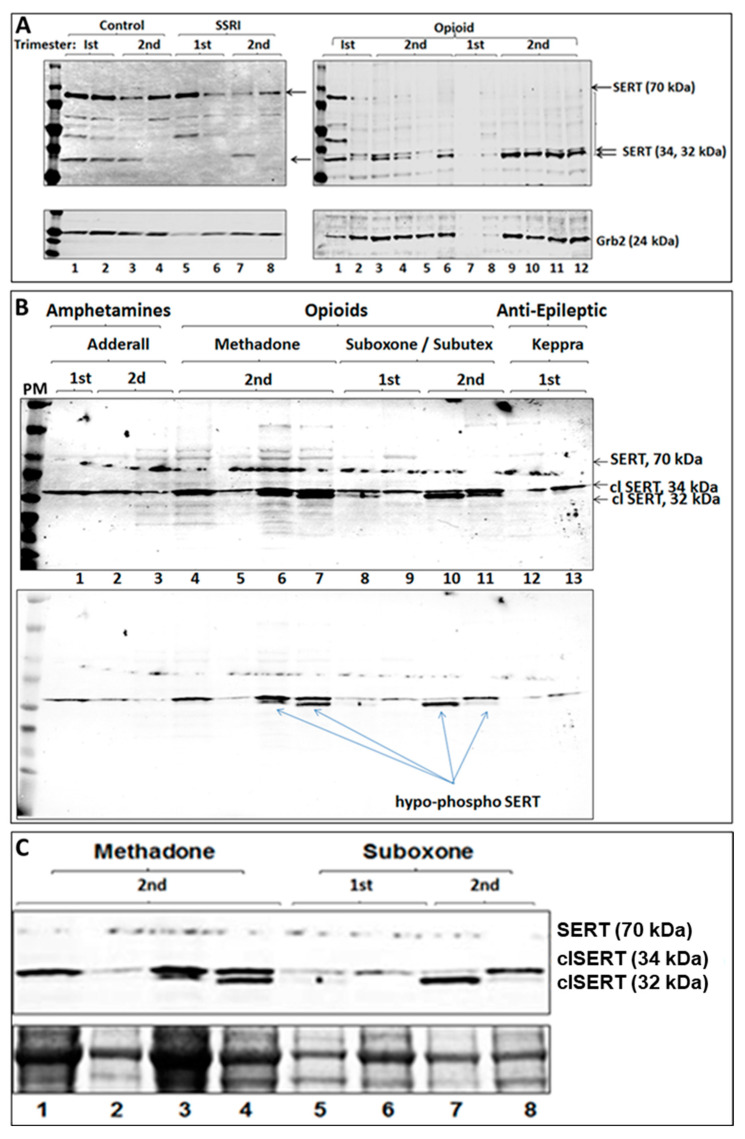
**SERT is modified by maternal opioid exposure.** (**A**) Immunoblot detection of SERT in placental vesicles of first and second trimester fetuses, using a rabbit polyclonal anti-serotonin transporter antibody specific for 70 kDa SERT. Grb2 served as a loading control. Subjects who admitted to chronic opioid use were compared to fetal sex-, GA-, and maternal age-matched controls. **Left panel**, lanes 1–4 are unexposed controls. Lanes 5–8 are subjects not exposed to opioids or EtOH but who were being treated for depression with SSRIs. **Right panel**, 12 opiod-exposed cases not exposed to other drugs. (**B**) Alteration of the SERT protein band pattern is seen only in opioid-exposed cases (lanes 4–10), not with other drug exposures—Adderall (lanes 1–3) and Keppra (lanes 11–12). The cleaved forms of SERT were observed only in opiate cases (lanes 4–10, top panel). (**C**) The smaller of the double-bands between 34 and 32 kDA representing cleaved forms of SERT (clSERT) in the opioid-exposed cases may represent a hypo-phosphorylated SERT fragment (see Figure 3).

**Figure 3 ijms-25-11570-f003:**
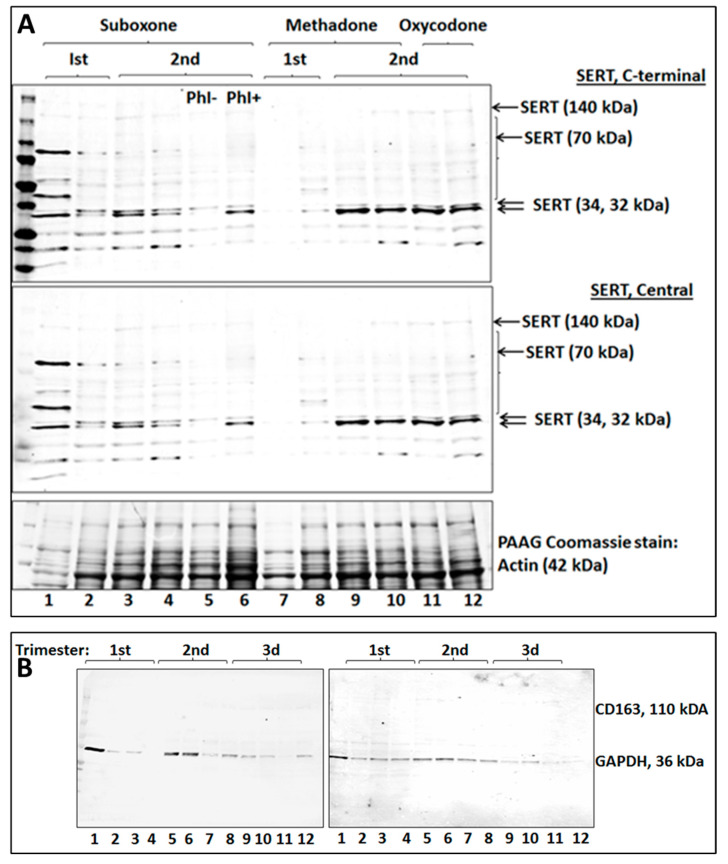
**Cleaved SERT isoforms in opioid-exposed cases are within the central domain**. (**A**) The immunoblot used for Figure 3 was used for detection of the central domain and C-terminal of SERT in placental vesicles from first and second trimester human pregnancies. Western immunoblots demonstrate expression of SERT isoforms in vesicles from 12-opioid exposed placentas at two GAs. Post-translational modification of SERT was studied by treatment with phosphatase inhibitors (Lane 5 vs. 6). This resulted in enhancement of the lower of the opioid-induced bands at 32–34 kDa, suggesting that most of this smaller clSERT is unphosphorylated. Antibodies to the N-terminal of SERT did not recognize either of the smaller (32 and 34 kDa) bands, while an antibody to the C-terminal region of SERT recognized both bands (upper panel), identical to the bands identified by antibodies to the central region of SERT (lower panel). SDS-PAGE stained with Coomassie blue was used to assure equal protein loading. The actin band (42 kDa) is labeled. (**B**) An antibody specific for the macrophage marker CD163 was used as a negative control, in combination with an antibody to GAPDH as a positive control, to confirm the purity of placental vesicles, the lack of contamination by other sources of SERT, and the specificity of the SERT double bands in panels (**A**,**B**) Note the absence of a band at 70 kDa.

**Figure 4 ijms-25-11570-f004:**
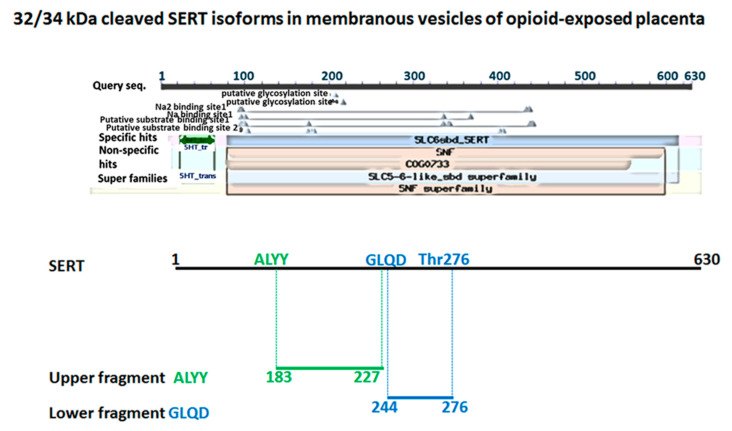
**SERT sequencing in opioid cases.** The Edman degradation experiment was performed to get the N-terminal sequence of truncated SERT protein double-band versions in opioid-exposed placental vesicles. Two SERT fragments were present in the opioid samples. The sequences were compared with two 5HT transporter isoforms downloaded from Unipro. Sequence alignment was performed using ClustalW2 on two isoforms of SERT fragments on SLC6A4 (Synonyms:HTT, SERT; Organism Homo sapiens (Human), Taxonomic identifier9606 [NCBI]). The first four amino acids of the Central region of SERT were detected in two potential truncated bands corresponding to the sequence of SERT. In combination with the western blot results, we confirm the presence of a truncated SERT protein. Truncated proteins were located using the sequence alignment tool. Version 1 (the upper band) encompasses the amino acids 183–227 (out of 672) of human SERT. Truncated Version 2 (lower band) encompasses amino acids 224–276 of human SERT, which includes the phosphorylation site Thr276.

**Figure 5 ijms-25-11570-f005:**
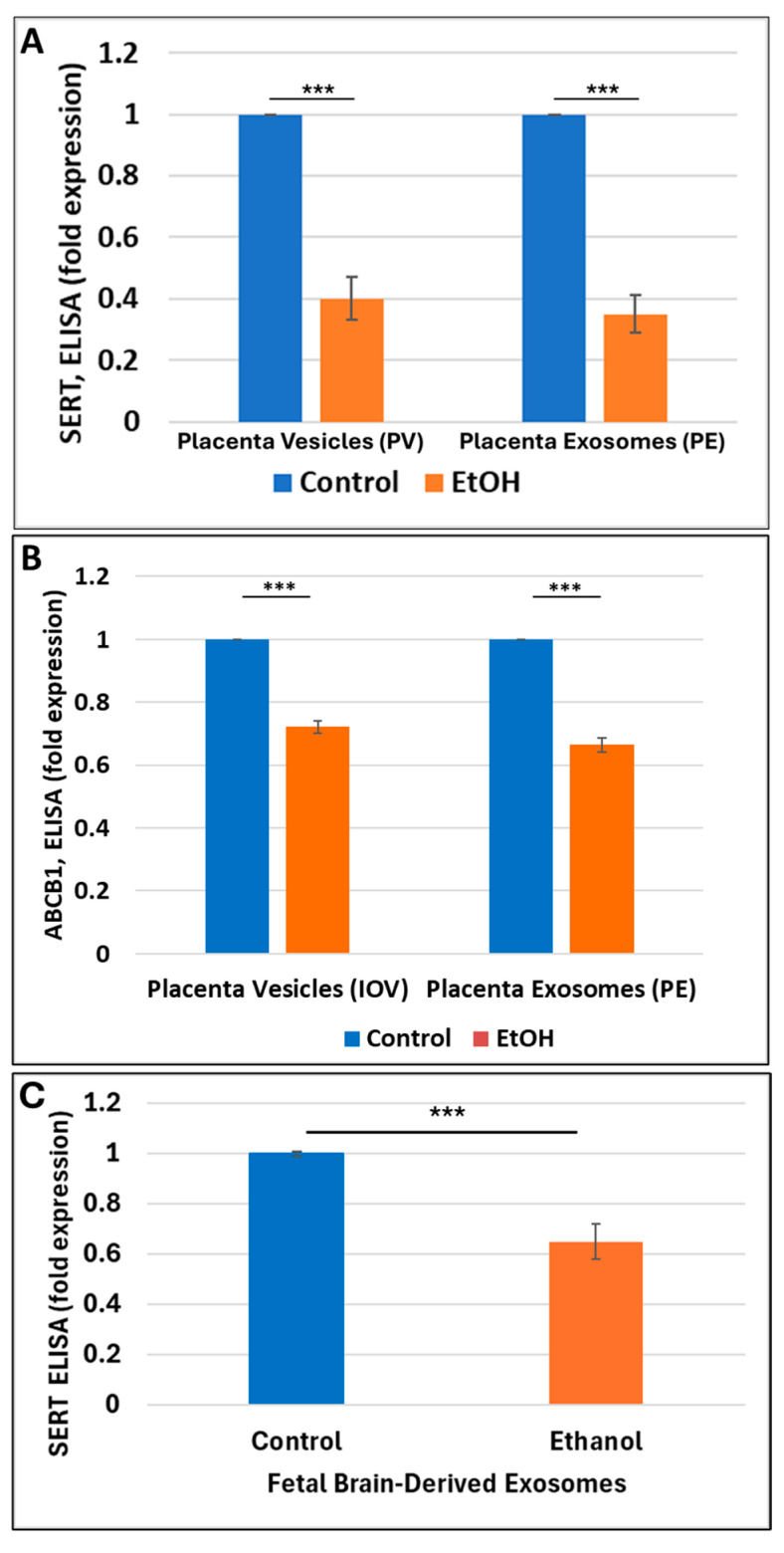
**Prenatal alcohol exposure-associated downregulation of SERT and ABCB1 in placenta vesicles, placenta-derived exosomes, and fetal brain-derived exosomes.** (**A**) EtOH exposure is associated with downregulation of total SERT in right side out (brush border membranous) placental vesicles (IOV) and placenta-derived exosomes from maternal blood (PE; bars 3–4) from human placenta tissues (n = 20) obtained from first and second trimester pregnancies. Analysis was completed by ELISA using a SERT ELISA kit, and results were obtained in picograms/microliters according to SERT standards. Bars represent fold changes between SERT levels in vesicles from EtOH-exposed cases compared to unexposed controls. (**B**) A similar reduction in expression of ABCB1 was seen in inside-out vesicles IOV. (**C**) Downregulation of SERT in fetal brain-derived exosomes. All assays were completed in triplicate (catalog #HS0096, detection range is within 0, 0.5, 1.0, 2.5, 5.0, 10 ng/mL, or picogram/microliter, and sensitivity is 0.1 ng/mL). ( *** for *p* < 0.001).

**Table 1 ijms-25-11570-t001:** Maternal blood and placenta tissues used in studies. **A**. Placental tissue samples and maternal blood samples from EtOH-exposed cases (n = 20) and controls (n = 20) were matched individually for fetal sex, GA, and maternal age. PCR for the SRY gene on the Y chromosome for sex determination was performed for EtOH cases and control samples (see 4, 5). **B**. Medication exposure groups for most experiments using qWestern blot and ELISA assays for placenta and blood tissues (n = 11 total) vs. controls (n = 20) (see 5, 15). Individual matching of controls for maternal and gestational age was as in **A**, but the ages are not included because of the small number and drug-type heterogeneity of subjects. **C**. Clinical characteristics of subjects by opioid exposure (n = 20) vs. controls (n = 20) (see 5, 11).

**A**		
	EtOH-consuming subjects (n = 20)	Control subjects (no EtOH, n = 20)
Maternal Age (years ± SD)	26.17 ± 2.15	22.34 ± 1.70
Gestational Age (weeks ± SD)	15.47 ± 1.33	15.16 ± 1.42
**B**		
	Drug/Medication	Subjects (number)
Control	None	11
SSRI	Celexa, Lexapro, Sertraline	5
Amphetamine	Adderall	4
Anti-Epileptic	Keppra	2
**C**		
	Opioid consuming subjects: methadone, suboxone, oxycodone (n = 20)	Control subjects (n = 20)
Maternal Age (years ± SD)	27.01 ± 3.36	24.8 ± 5.4
Gestational Age (weeks ± SD)	15.3 ± 3.2	14.8 ± 2.8

**Table 2 ijms-25-11570-t002:** List of laboratory procedures described in Materials and Methods.

	Methods
1	Preparation of brush border membrane vesicles: ROV, IOV
2	Placenta-derived exosomes
3	Fetal brain-derived exosomes
4	Alkaline phosphatase (ALP) activity assay
5	qWestern-blot assay
8	ELISA
9	Protein sequencing

## Data Availability

This study collected demographic, behavioral, and laboratory data from normal, healthy women and from women who drank alcohol during pregnancy. Our research team supports all these activities and has developed a data-sharing plan. We also recognize that additional benefits from data sharing may arise in the future that are not apparent at this time, and we are prepared to work specifically with NIH in addressing all requests for raw data. At the present time, we have not deposited any of these raw data in an existing databank but will make the data available to other investigators on request, in a manner consistent with NIH guidelines. Consistent with NIH policy, shared data will be rendered “free of identifiers that would permit linkages to individual research participants and variables that could lead to deductive disclosure of the identity of individual subjects” Intellectual property and data generated under this project will be administered in accordance with both University and NIH policies, including the NIH Data Sharing Policy and Implementation Guidance of 5 March 2003, and 0925-0001 and 0925-0002 (Rev 07/2022 through 01/31/2026). With this caveat observed, data will be made available to the NIH/NICHD/NIAAA. Sufficient identifiers will be provided to the NIH so that research participants can be assigned a Global Unique Identifier (GUID), which is a universal subject ID that protects personally identifiable information (PII). Using the GUID, NDAR can bring together multiple types of data collected from a single participant, regardless of where and when those data were collected. Biological samples (blood, serum, exosomes, and RNAs) and data that are shared will be completely free of identifiers that would permit linkages to individual research participants. We will make biological samples, deidentified data, and associated documentation available to users only under a data-sharing agreement that provides for (1) a commitment to using the data only for research purposes, (2) a commitment to securing the data using appropriate computer technology; and (3) a commitment to destroying or returning remaining samples after analyses are completed. Intellectual property and data generated under this project will be administered in accordance with both University and NIH policies, including the NIH Data Sharing Policy and Implementation Guidance of 5 March 2003. As the FAIR data bank receives approval from the NIH, the data will be made available to that group as well. The NIH will be implementing a new specific policy regarding data sharing https://grants.nih.gov/grants/guide/notice-files/NOT-OD-21-014.html, as of 25 January 2023. We will adopt that policy also. Data will be also available at https://www.mdpi.com/ethics (accessed on 1 January 2025).

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
