# Peer review of "Prenatal Opioid and Alcohol Exposures: Association with Altered Placental Serotonin Transporter Structure and/or Expression"

_ijms, 2024, doi:10.3390/ijms252111570_

Round 1

Reviewer 1 Report

Comments and Suggestions for Authors

The research topic is highly significant, of broad interest, and forward-looking in the field of non-invasive fetal monitoring and diagnostics. Certainly, the study of exosomes has emerging interests that open an exciting window into placental and fetal molecular pathophysiology. With all of that, the study design and data interpretation are challenging, in part due to its inherent nature related to human studies, but equally or even to a greater extent, confusion related to presentation details.

1) A methods flow chart would be very helpful

2) Controls used to "normalize" data vary from experiment to experiment--Grb2, control [whatever that is], GAPDH, coomassie blue. Unfortunately, the direct control results show high variability in loading and the use of Grb2 and GAPDH could be problematic because the levels shift with oxidative stress. Ethanol or opioid-exposed placentas sustain stress responses which may account for apparent variation in sample loading.

3) Related to #2, Fig 1 uses two different measures of normalizing SERT and ABCB1—uniform data analysis would help with the interpretation—or justification for the switching.

4) Terminology pertaining to which samples were evaluated is very confusing.  Studies of Placental membrane vesicles, Serum-derived placental exosomes, and Serum-derived exosomes Fetal brain exosomes should be clearly delineated. Fetal brain exosomes are not referred to in the Figures--perhaps this is one of the confusing points, or else the data were not shown.

5) Continuing from #4, Figures 2 and 3 show results of placental vesicles—presumably these correspond to placental membrane vesicles from the tissue and not exosomes. Comparisons with same-source placental tissue would help assess whether the vesicles represent tissue expression or aberrant tissue release of vesicles in relation to ethanol or drugs. In other words, what’s happening in the tissue. Also, direct comparisons with placental exosomes isolated from serum would strengthen the potential diagnostic approach.

6) Fig 2—lowest panel, probably Grb2, is not labeled

7) For the WBs, how was the original sample loading determined? Protein content, membrane vesicle content? This needs to be stated in the legends.

8) What is the rationale for using Grb2 as a loading control—is it known to be expressed in MVs? Please provide the citation.

9) In Fig 3, the SERT central and Sert C-terminal blots look identical—is this a duplication error?

10) CD163 immunoreactivity is not easily seen to confirm specificity of the MV isolation. Control studies using the same antibodies with placental tissue should be done since MVs come from the placenta and should, therefore, be positive.

11) The difficulty obtaining adequate clinical exposure data is appreciated. Is there any way to stratify exposure doses with outcomes? Information about binge versus chronic alcohol during pregnancy and pre-gestation would be pertinent. Also Opioid use pre-pregnancy information would be of interest in relation to DNA damage and sequencing results related to SERT cleavage.

12) How was sample integrity ensured? Timing in relation to pregnancy termination and harvest appeared systematic. However, the methods of pregnancy termination could impact the integrity of placental and fetal tissue.  Were the pregnancy termination methods uniform? Comparisons with preterm delivery placentas could help resolve these matters.

13) Were there any subject/sample inclusion/exclusion criteria?

14) Was the maternal blood obtained at the time of delivery?

15) Placental membrane vesicles vs EVs: how was cross-contamination from maternal blood prevented?  Concerns relate to the nearly identical results from placental MVs and Serum EVs (from maternal serum).

16) Nanoparticle profile comparisons of etoh vs opioid vs control would be of interest

17) Fetal brain exosome data are not shown [stated as negative and an indication of placental EV purity in previous publications]. If they are, then the comments about the confusing manner of data presentation are reinforced.  Fetal brain exosome data need to be labeled in association with the blots and ELISA results. It appears that comparisons were made with placental membrane vesicles and placenta-derived serum exosomes. The legends do not include fetal brain exosomes, yet they are mentioned in the first line of the Discussion.

18) Check for typos, e.g. lines 231 and 234

Reviewer 2 Report

Comments and Suggestions for Authors

In the submitted research article the authors demonstrate that prenatal opioid exposure but not EtOH is associated with novel serotonin (5HT) transporter (SERT) isophorms as a consequence of  post-translatinal phosphorylation of SERT cleavage products. In general, the manuscript is well stuctured, the experimental design is good and presented data support the conclusions of the study. However this reviewer suggest few improvements in introduction, methods and figure/figure legends to add clarity.

This reviewer does not find suitable to a research article the division of introduction into sub-paragraphs.

Figure 1: insert representative figures for Western blot and Coomassie blu staining, better explain normalization procedure (i.e., insert housekeeping protein and reference sample) and how data are reported in graphs (i.e., panel A, what do the authors mean for fold (i.e. fold increase vs?)? Panel B, what about the houekeeping and why date are reported normalized vs control?).

Figure 5, better explain the meaning of ELISA (fold expression)

In Elisa assay inclure detection limits, replicates, inter-intra coefficient of variability

537-546: Western blot procedure needs more details (i.e., primary and secondary antisera dilutions). What is the rationale for 3 housekeekings?

Reference list needs format revision.

iThenticate score (54%) is quite high. In method section avoid repetition of elsewhere published full protocols: cite the original source and focus on the specific changes done in this study
